# The Problem with Inventing Molecular Mechanisms to Fit Thermodynamic Equations of Muscle

**DOI:** 10.3390/ijms242015439

**Published:** 2023-10-22

**Authors:** Josh Baker

**Affiliations:** School of Medicine, University of Nevada, Reno, Reno, NV 89557, USA; jebaker@unr.edu

**Keywords:** thermodynamics, muscle, myosin, actin, motor, force, mechanics, kinetics, free energy, paradigm

## Abstract

Almost every model of muscle contraction in the literature to date is a molecular power stroke model, even though this corpuscular mechanism is opposed by centuries of science, by 85 years of unrefuted evidence that muscle is a thermodynamic system, and by a quarter century of direct observations that the molecular mechanism of muscle contraction is a molecular switch, not a molecular power stroke. An ensemble of molecular switches is a binary mechanical thermodynamic system from which A.V. Hill’s muscle force–velocity relationship is directly derived, where Hill’s parameter *a* is the internal force against which unloaded muscle shortens, and Hill’s parameter *b* is the product of the switch displacement, *d*, and the actin–myosin ATPase rate. Ignoring this model and the centuries of thermodynamics that preceded it, corpuscularians continue to develop molecular power stroke models, adding to their 65-year jumble of “new”, “innovative”, and “unconventional” molecular mechanisms for Hill’s *a* and *b* parameters, none of which resemble the underlying physical chemistry. Remarkably, the corpuscularian community holds the thermodynamicist to account for these discrepancies, which, as outlined here, I have done for 25 years. It is long past time for corpuscularians to be held accountable for their mechanisms, which by all accounts have no foundation in science. The stakes are high. Molecular power stroke models are widely used in research and in clinical decision-making and have, for over half a century, muddied our understanding of the inner workings of one of the most efficient and clean-burning machines on the planet. It is problematic that corpuscularians present these models to stakeholders as science when in fact corpuscularians have been actively defending these models against science for decades. The path forward for scientists is to stop baseless rejections of muscle thermodynamics and to begin testing corpuscular and thermodynamic mechanisms with the goal of disproving one or the other of these hypotheses.

## 1. History of System Mechanics

To quantitate the mechanics of a system of molecules, force must be defined and be definable on some scale. Specifically, either the force of individual molecules in the system, *F_uni_*, is defined if definable and used to determine the system force, *F*, or the system force, *F*, is defined as measured and used to determine the average molecular force. The scale at which force is definable is not a choice made by the observer.

If one measures the force of a molecule on a time scale much longer than the time scale at which the force of that molecule thermally fluctuates, only the average force of that molecule is definable. If the dynamic force of that molecule is transmitted to (distributed among) other neighboring molecules, the average force on that same time scale is not definable for any one molecule independent of another. If these forces are distributed among more than three thermally fluctuating molecules, this many-body problem becomes nearly impossible to solve, and at best, only an average force of the collection of molecules can be defined. This becomes an impossible problem to solve if those molecules exchange forces with an unconstrained, thermally fluctuating universe.

This problem is dramatically simplified by a container that constrains and defines the surroundings of a system of molecules, in which case a measurable force, *F*, exerted on a system of molecules constrains the mechanics of the molecules in that system. In this case, *F* is a definable model parameter that determines the average molecular force and energetics of the molecules in the system. This is fundamentally different from defining molecular forces, *F_uni_*, as mechanically constrained, definable model parameters that determine the system force, *F*. The former is referred to as thermodynamics, and the latter is referred to as corpuscular mechanics.

Figure 1 shows two timelines illustrating the history and distinction between corpuscular mechanic (top) and thermodynamic (bottom) descriptions of a system of molecules held at a constant force, *F*. For a system that changes its state on a time scale much slower than the time scale over which forces are dynamically distributed, corpuscular mechanics does not apply because on this time scale *F* determines an average *F_uni_* not the other way around. Here, the state of the system can only be defined in terms of a molecular ensemble (not molecules) constrained by *F* (not *F_uni_*). The energetic details that are lost with the definition of a molecular ensemble are explicitly accounted for by a change in entropy, TΔS, of the system.

Robert Boyle was a proponent of the corpuscularian philosophy. He wrote, “the phenomena of the world are physically produced by the mechanical properties of the parts of matter and that they operate upon one another according to mechanical laws. It is of this kind of corpuscular philosophy that I speak”. Today, the term “corpuscular mechanics” is rarely used because it is widely considered an obsolete scientific philosophy since thermal fluctuations and entropy preclude Boyle’s molecular determinism. Nevertheless, the term still describes the philosophy and computational models in which the force of a system is defined by the sum of the forces of its mechanical parts, absent thermal fluctuations and entropy, justifying its use herein.

In 1662, at the dawn of chemistry, Boyle observed that the pressure of a gas increases linearly with decreasing volume [1]. Boyle proposed that pressure increases because air molecules, consisting of tiny, coiled springs, are compressed by the decrease in volume (Figure 1, coiled springs adapted from Boyle’s drawings [1]; cylinder adapted from Carnot’s drawings [2]). Here, the pressure of a gas (Figure 1, *F* exerted on a piston) is determined from the forces, *F_uni_*, of imagined springs of air summed over all springs. Interestingly, Hooke, who at the time was studying the relationship between the force and displacement of springs, worked with Boyle to develop the mathematics for this model.

In 1690, Amontons observed that the pressure of a gas increases with temperature—an observation that inspired Amontons to create the first heat engine by heating a gas and then allowing it to expand [3]. Amontons proposed a corpuscular mechanic model for his heat engine in which a heat molecule, referred to at the time as a phlogiston, tensions a spring of air to increase its outward force, *F_uni_*, after which the tensioned spring relaxes through a molecular power stroke upon expansion of the gas. According to Amontons, the transfer of heat to mechanical work occurs within individual air molecules.

Amontons’ corpuscular mechanic model for a heat engine is directly analogous to the corpuscular mechanic model for muscle contraction proposed by Huxley more than 250 years later. According to Huxley, muscle force, *F*, is determined from the well-defined forces of individual springs of myosin, *F_uni_*, summed over all myosin springs. The corpuscular mechanism of muscle contraction is that a single ATP molecule (through a hydrolysis reaction) tensions the spring of a single myosin molecule, after which the tensioned spring of myosin relaxes through a molecular power stroke upon muscle shortening. Here, consistent with Boyle and Amontons, the transfer of chemical energy to mechanical work occurs within individual myosin molecules.

In 1730, Bernoulli proposed that gas pressure results not from the force, *F_uni_*, of corpuscular springs of air but from thermally diffusing gas molecules that generate an outward force when they strike the walls of a cylinder. Through his kinetic theory of gases, Bernoulli showed that temperature determines pressure and that the conduit for this system-level relationship is not individual gas molecules but the kinetic energy of an ensemble of gas molecules that is determined by temperature and defines pressure (*F* in Figure 1).

In 1824, with the advent of thermodynamics, Carnot used mathematical relationships between system parameters T, P, and V in describing a work loop for an idealized thermodynamic system [2]. He demonstrated that the work performed by the expansion of a gas occurs through a thermodynamic power stroke (an ensemble mechanism), not a molecular power stroke. Through a change in what would later be referred to as system entropy, ΔS, he demonstrated a system-level conduit for energy transduction, disproving Amontons’ molecular power stroke mechanism for the work performed by a heat engine.

In 1850, Gibbs showed that reaction-free energies (chemical forces), defined in part by a reaction entropy, ΔS, determine the energy available for work by a chemical reaction at constants T, P, and F, demonstrating that energy transduction through chemical reactions occurs through a molecular ensemble conduit, not through individual molecules, disproving molecular power stroke mechanisms in chemistry. In describing the dynamic basis (the many-body problem above) for why corpuscular mechanics cannot be defined, Gibbs wrote, “The laws of thermodynamics… express the laws of mechanics for… systems as they appear to beings who have not the fineness of perception to enable them to appreciate quantities of the order of magnitude of those which relate to single particles” [4]. In describing the entropic basis for why corpuscular mechanics cannot be defined, Gibbs wrote, “If we wish to find in rational mechanics an a priori foundation for the principles of thermodynamics, we must seek mechanical definitions of temperature and entropy” [4].

In 1938, A.V. Hill made measurements of heat and power output by shortening muscle and showed that chemical forces (chemical free energies) decrease linearly with muscle force, *F*. He also showed that the difference between these forces is proportional (with a proportionality constant, *b*) to the energy available for mechanical work by shortening muscle. In other words, A.V. Hill established muscle as a thermodynamic system [5]. Although little was known at the time about muscle proteins or even ATP, Hill’s results demonstrated that the energy-transducing chemistry in muscle occurs through an ensemble conduit, not through molecular power strokes. Hill’s work established “a framework into which (the chemists’) detailed machinery must be fitted” [6] not thermodynamic parameters (*a* and *b*) to be fitted by corpuscular mechanics.

In 1957, Huxley proposed a corpuscular mechanic model of muscle contraction, defining both chemical and mechanical forces at the level of a single myosin motor and describing a single myosin motor as the conduit for free energy transduction. That is, the free energy for the hydrolysis of a single ATP molecule is transferred to a single myosin motor, which then transfers that energy to mechanical work through the power stroke of that motor [7]. The only possible justification for this departure from thermodynamics is that a crystalline lattice within muscle mechanically constrains molecular forces, *F_uni_*. In 1974, T.L Hill established the formal requirements for this model. However, it has since become clear that muscle does not contain a crystalline lattice and does not meet Hill’s formal requirements [8,9,10,11,12,13,14,15,16]. Nevertheless, almost every model of muscle contraction to date is a corpuscular mechanic model. Absent any formal justification, corpuscular mechanic mechanisms continue to be defined and refined with the assumption that molecular forces crafted by unconstrained human intuition are more deterministic of the mechanical properties of muscle than the measurable force, *F*, physically exerted on it.

In 1998, after directly observing that the molecular mechanism of muscle contraction is a mechanical switch [9,11,17] and not a molecular power stroke, we established the chemical basis for A.V. Hill’s thermodynamic muscle model and have since fully developed a binary mechanical model of muscle contraction that accounts for many mechanical and energetic aspects of muscle contraction [9,10,15,18]. In this model, the conduit for energy transduction is the ensemble actin–myosin ATPase reaction, within which the mechanical switch is an intermediate chemical step. The system reaction-free energy drives the ensemble reaction, and an ensemble of molecular switches performs work through a thermodynamic power stroke. The entropic force, ΔS, of a binary mechanical system [15,19] disproves molecular power stroke models of muscle contraction.

## 2. Muscle Thermodynamics

Muscle is a dynamic macromolecular assembly of proteins contained in a cell (a muscle fiber) within which every protein in the assembly is compliant and undergoes thermal fluctuations on a femtosecond time scale. While it might be possible to define corpuscular mechanics on a time scale shorter than this [20], it is not possible to define corpuscular mechanics on the millisecond time scale of enzyme-catalyzed reactions. In this case, only the chemistry of molecular ensembles (not molecules) can be defined, where the energetic changes lost through the definition of a molecular ensemble are accounted for by TΔS. Because a hybrid between corpuscular mechanics (molecules with no entropic changes) and thermodynamics (molecular ensembles with entropic changes) is not defined, models that proposes a link between these theories [21,22,23] assume that the mechanical state of a molecule is well-defined (i.e., is not a time averaged molecular force defined by the system force, *F*) on the time scale of an enzyme-catalyzed reaction.

The mechanical state of every protein in muscle stochastically changes millions of times over the time scale of catalyzed reactions. As a result, during a catalyzed reaction, force and energy are not well defined in any one part of the assembly because they are randomly and rapidly distributed throughout it. While molecular forces are not locally well defined on the timescale of chemical reactions, the macroscopic force exerted on a muscle fiber is. Muscle force, *F*, equilibrates with the complex, dynamic network of protein springs in muscle, providing a measurable, well-defined, macroscopic constraint on the mechanochemistry of the dynamic protein ensembles contained within the muscle system [10,24,25].

This is the physical foundation for chemical thermodynamics, where ensemble chemistry and mechanics are defined in terms of macroscopic system constraints [26], like pressure, temperature, and here, muscle force. If muscle is a chemical thermodynamic system, we would expect system-free energies and rate constants to be defined by muscle force *F*. In fact, in 1938, based on careful measurements of the relationship between muscle force and muscle energetics, A.V. Hill observed that the energy available for work by shortening muscle (i.e., the system-free energy) decreases linearly with *F*, establishing muscle as a thermodynamic system and defining the muscle equation of state [5]. Incorporating this equation into a statement of energy conservation, Hill derived the muscle force–velocity relationship [5]. This thermodynamic relationship emerges from the impossibly complex molecular mechanics within the system and thus is not a relationship that can be determined from well-defined molecular mechanics (e.g., *F_uni_* and molecular power stroke mechanisms).

In 2000, after the molecular mechanism of muscle contraction was directly observed in both muscle [9,10,16] and single-molecule mechanics studies [11,17] to be a molecular switch not a molecular power stroke, we established the mechanochemical basis for Hill’s thermodynamic equation of state [15,18]. Twenty years later, I have finished characterizing the basic mechanical performance (force generation, work loops, tension transients, and force–velocity) of a simple thermodynamic muscle model [15] and have shown that it aligns with many mechanical and energetic aspects of muscle contraction.

Specifically, induced by the formation of a strong bond to actin and gated by phosphate, P, release, a single myosin motor undergoes a discrete lever arm rotation that displaces an external compliant element, generating force proportional to the stiffness of the external element [9,10,11,15,27]. In muscle, this myosin switch displaces a dynamic network of molecular springs. While the work performed on individual springs is impossible to define because molecular forces are stochastically distributed throughout muscle on a time scale much faster than that of the switch, it is trivial to determine the work performed on a system held at a force, *F*. The work performed by a myosin switch on a system of springs equilibrated with the system force, *F*, is proportional to *F* [10,16]. With its reversal, the work performed by the system on a switch is also proportional to *F*. This means that the binding-free energy that drives the switch, along with its forward and reverse rate constants, are all explicit functions of *F*. This binary mechanical model [15] is the mechanochemical basis for A.V. Hill’s muscle equation. The energy available for work—the actin–myosin binding-free energy—decreases linearly with *F* [18].

## 3. Muscle Corpuscular Mechanics

While it is hardly a radical claim that thermodynamics applies to muscle, a thermodynamic theory of muscle contraction is in opposition to Huxley’s 1957 theory of muscle contraction [7], which is based on a pre-thermodynamic corpuscular mechanic philosophy (Figure 1). Huxley’s 1957 molecular power stroke model is directly analogous to the molecular power stroke model of a heat engine proposed by Amontons in the 1600s. In both cases, a molecular form of energy (heat or chemical) is transferred to a molecule, tensioning a molecular spring to generate force. The molecular spring then relaxes through a molecular power stroke to perform work. According to this model, individual molecules are the conduits for energy transduction.

In 1957, Huxley considered muscle to be a crystalline lattice, and so at the time—if one ignored the work of A.V. Hill—a corpuscular mechanic model was reasonably justifiable. However, today, we know that all proteins in muscle are compliant and thermally fluctuate. Bernoulli’s discovery that air molecules thermally fluctuate (the kinetic theory of gas) eventually led to the thermodynamic equation of state for a gas (the ideal gas law), which superseded the springs of air as the mechanism for Boyle’s pressure-volume relationship. It also led to the Carnot cycle with a thermodynamic power stroke mechanism that superseded Amontons’ molecular power stroke. Gibbs’ understanding that molecules thermally fluctuate led to the thermodynamic equation of state for chemical reactions (the Gibbs free energy equation), which supersedes mass action. A.V. Hills’ understanding that the chemical components of muscle thermally fluctuate led to Hill’s thermodynamic equation for muscle (Hill’s force–velocity relationship), which supersedes the springs of myosin as the mechanism for muscle contraction. In all cases, as Gibbs described, the laws of thermodynamics (equations of state) express the laws of mechanics for a system of molecules as measured, not the laws of mechanics of molecules as we imagine them to be.

The scientific opposition to corpuscular mechanics is overwhelming. As Einstein stated, thermodynamics “is the only physical theory of universal content, which I am convinced, that within the framework of applicability of its basic concepts will never be overthrown”. The experimental evidence against a corpuscular mechanic model of muscle is likewise significant. A corpuscular model of muscle requires that *F_uni_* be definable, which means that it must be constrained by a crystalline lattice, yet today it is well established that muscle proteins are compliant and thermally fluctuate [8]. A corpuscular model of muscle requires that a spring of myosin be stretched to generate internal force before it relaxes to generate movement through a continuous corpuscular power stroke, yet myosin motors are observed to generate both external force and movement through one discrete molecular switch [9,10,11,15]. A corpuscular model requires that myosin motor kinetics and energetics are defined independently of muscle force, yet myosin motor kinetics and energetics vary linearly with muscle force [5,10].

Throughout scientific history (Figure 1), even in a post-thermodynamic era, corpuscular mechanics keeps returning as a prevailing hypothesis. For example, chemists continue to describe molecules in a specific chemical state as being physically pushed out of that state (mass action) by chemical forces exerted by the mere presence of molecules in that state (chemical activities) rather than being pulled by entropy [28]. In biology, a single ATP molecule is described as transferring corpuscular chemical energy to a single protein, which is directly analogous to a single phlogiston molecule transferring corpuscular heat energy to a single air molecule. In physiology, for over 65 years, an entire scientific community has been unquestioning in its defense of molecular power strokes against thermodynamic power strokes.

From a science of science perspective, this corpuscular mechanic conviction (or perhaps a thermodynamic aversion) is a stark illustration of the problematic difference between science and the humanness of scientists (Kuhn’s natural science). The solution to this problem is to better train scientists to not trust their corpuscular intuition and to incorporate into their scientific toolkit an understanding of thermodynamic mechanisms. In molecular and cellular biology, this requires a reversal of the mindset that the more we learn about structural details, the more we will know about mechanisms. Somehow, this is less the mindset in other scientific disciplines. We do not attempt to understand how an engine works by studying the mechanics of steel or gas molecules. We do not attempt to understand fluid flow by studying the forces exerted by water molecules. We do not attempt to understand the transmission of signals through a fiber optic in terms of energy transduction through silica molecules. Biological functions are similarly large-scale phenomena, yet in biology, these phenomena are often projected on to smaller scales as behaviors that belong to proteins and cells. In other words, these structures are defined as the corpuscular conduits through which energy transduction (i.e., function) occurs.

The essential question becomes at what scale is biological function (i.e., energy transduction) physically contained? A single protein, when physically constrained in a laboratory [17,29], might be a well-defined conduit for energy (i.e., have a well-defined average force and energy) but when functioning as part of a larger dynamic cellular environment, becomes part of a much larger conduit for energy transduction. In muscle, macroscopic muscle force provides a more fundamental constraint on force-generating chemistry than do imagined molecular constraints. On a larger scale, Pang et al. recently provided experimental support for a model of brain activity that implies “the geometry of the brain may represent a more fundamental constraint on (neuronal) dynamics than complex interregional connectivity [between cells]”, implying that in the brain the conduit is much larger than even a network of cells [30].

While necessary for advancing biological sciences, training scientists to ignore their corpuscular intuition is a challenge. When I explain to students Boyle’s springs-of-air interpretation for his pressure-volume relationship, I am on occasion asked, “Why isn’t it taught that way? That makes so much more sense to me”. Similarly, when I challenge muscle biophysicists on their springs-of-myosin interpretation of muscle force, their final defense is often, “It’s the way that I like to think about it”. The problem is that it is much more than that. It is a disregard for science. For 25 years, my thermodynamic arguments have been consistently countered by an argument considered damning by the corpuscularian community—that thermodynamics (e.g., a thermodynamic power stroke) is inconsistent with corpuscular mechanics (e.g., a molecular power stroke). While certainly true, the many peer reviewers who have passionately made this argument and the editors who agree with them seem unaware and, despite my best attempts, remain unconvinced that this is a challenge to corpuscular mechanics, not thermodynamics.

## 4. Thermodynamics Is Inconsistent with Corpuscular Mechanics

Despite two centuries of scientific challenges to corpuscular mechanics, 85 years of unrefuted evidence that muscle is a thermodynamic system [9,10,11,12,13,14,15,16,24,25], and a quarter century of direct observations that the molecular mechanism of muscle contraction is a molecular switch, not a molecular power stroke, almost every model of muscle contraction in the literature to date is a molecular power stroke model. These models describe unconstrained in silico forces generated through molecular mechanisms that are crafted by human imagination and computationally endowed with the power to overcome thermal fluctuations and system entropy to become more deterministic of the mechanical properties of muscle than muscle force, *F*, itself. These simulations are justified solely by the argument that “it’s the way I like to think about it”.

Two of the most important parameters describing muscle mechanics are the thermodynamic parameters *a* and *b* in Hill’s force–velocity relationship. Because the force–velocity relationship is derived directly from a simple binary thermodynamic model, these are well-defined model parameters. Specifically, parameter *a* is the internal force against which unloaded muscle shortens, and *b* is the product of the myosin step size, *d*, and the actin–myosin ATPase rate [15,18]. Ignoring this thermodynamic model and centuries of thermodynamics preceding it, corpuscularians have for over 65 years created one new molecular power stroke model after another, each describing different mechanisms for *a* and *b*, none of which are consistent with the basic underlying physical chemistry (most recently [31]). No attempts are made to test one proposed corpuscular mechanism against another because the strain-dependent corpuscular kinetics arbitrarily defined to fit muscle force–velocity curves are non-physical and will never be observed experimentally. Yet, the multitude of different mechanisms proposed over the past 65 years somehow all purportedly support and never challenge corpuscularianism with each new proposed mechanism considered a rediscovery of corpuscularianism as recently described in “Hypothesis: Single Actomyosin Properties Account for Ensemble Behavior in Active Muscle Shortening and Isometric Contraction” [32]. Although different corpuscularians propose completely different mechanisms for *a* and *b*, they all view their models as being consistent with the model for *a* and *b* proposed by Huxley in 1957. Clearly, the goal of the corpuscularian community is more to support a philosophy than it is to determine mechanisms for muscle contraction (e.g., for *a* and *b*).

While they do not hold each other to account for differences between proposed mechanisms, the corpuscularian community has for over 25 years held the thermodynamicist to account for proposed mechanisms, which for over 25 years I have done. Table 1 summarizes different mechanical properties of muscle along with the disparate underlying mechanisms described by corpuscular mechanics and thermodynamics. Over the past 25 years, I have provided theoretical and experimental evidence supporting the thermodynamic mechanisms (Table 1 references), while the corresponding corpuscular mechanisms remain largely unsupported in the literature. After 65 years, it is long past time for the corpuscularian community to, first, come to a consensus on mechanisms for the most basic mechanical properties of muscle, such as *a* and *b*, and second, provide experimental and theoretical support for those mechanisms.

In the face of what is becoming insurmountable evidence against corpuscular mechanic models of muscle, the corpuscularian community continues to reject thermodynamic models, only now based on the argument that because no model is perfect, any model will do, and so a thermodynamic model is not needed. By claiming that thermodynamics and one’s imagination (“it’s the way I like to think about it”) are equally justifiable as foundations for a model, these reviewers are as a last resort abandoning science for their corpuscularian convictions. Molecular power stroke models of muscle contraction are widely used in research, drug discovery, and the discovery of disease mechanisms. If we as scientists tell physicians, researchers, patients, investors, and the public that we are using science to help solve these problems, when in fact the models we use are based on fanciful convictions challenged by science, that matters. Moreover, muscle is one of the most efficient, clean-burning machines on the planet, yet corpuscular mechanic models have muddied our understanding of how these machines work. As a result, we have gone decades without the extraordinary physical chemical insights that a thermodynamic analysis of muscle is now providing [19,28]—insights that have great potential to advance clean energy technologies. That, too, matters.

Throughout the history of science, corpuscular mechanic convictions among scientists have misled science and continue to do so today (Figure 1). From a science of science perspective, it is important that we better understand and address this human barrier to the advancement of science. With a focus on advancing muscle physiology, it is time to pause efforts at inventing new corpuscular mechanisms until, as scientists, we disprove one or the other of these theories. Do individual myosin molecules in muscle generate physically constrained molecular forces, *F_uni_*, that determine muscle force? Or does an ensemble of myosin molecules generate a system force, *F*, that determines average molecular forces? [9,11,15,17].

## Figures and Tables

**Figure 1 ijms-24-15439-f001:**
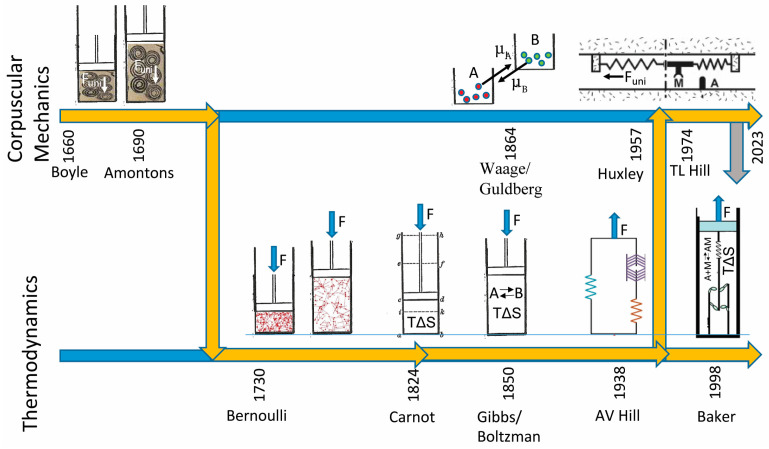
Two timelines for the history of both corpuscular mechanics (**top**) and thermodynamics (**bottom**). The top line illustrates the history of corpuscular mechanics, which is the obsolete 17th-century scientific hypothesis that posits that the mechanical properties of a system are defined by the mechanical properties of the molecules in that system. In other words, the force of the system, *F*, is well defined by the sum of well-defined molecular forces, *F_uni_*. The bottom line illustrates the history of thermodynamics, which is the theory that the force, *F*, exerted on a system mechanically constrains the mechanics and energetics of the molecules contained in that system. A hybrid between corpuscular mechanics (molecular states with no entropic changes) and thermodynamics (states of a molecular ensemble with entropic changes) is not defined, and so there is no formal link between these theories (gray arrow).

**Table 1 ijms-24-15439-t001:** Differences between corpuscular mechanics and thermodynamic models of muscle contraction.

Measurement	Corpuscular Mechanism *	Thermodynamic Mechanism
System State	Sum of molecular states	State of molecular ensemble
Molecular mechanism	Molecular power stroke	Molecular mechanical switch [9,11,15,17]
Isometric muscle force, *F_o_*	Determined by number of bound myosin motors, *N_bound_*	Determined by actin–myosin binding energy [10,33]
Muscle force, *F*	Determined by molecular states	Determines the state of the system [5,10,24]
Unloaded Shortening Velocity, *V_o_*	Not influenced by chemomechanical forces	Determined by chemomechanical forces [11,12,13,14,34,35,36]
Hill coefficient, *a*	Complex molecular mechanisms ^†^ (many discrepancies, none tested)	The internal force against which unloaded muscle shortens [5,14,15,18,34]
Hill coefficient, *b*	Complex molecular mechanisms ^†^ (many discrepancies, none tested)	The product of the myosin step size and the ATPase rate [5,18]
Work production and Force generation	Each phase is a different complex molecular mechanism ^†^ (many discrepancies, none tested)	All phases are well-defined combinations of two explicit thermodynamic processes [15,37,38]
System entropy	0	–k_B_Tln[(*N_bound_* + 1)/*N_unbound_*] [15,19,28]
Thermal Energy	Heat content of molecules	Heat content of system

* While under certain conditions, *N_bound_* increases with binding energy. In general, none of the above corpuscular mechanisms are supported experimentally. ^†^ These mechanisms are often shaped by strain-dependent rate constants that are arbitrarily defined with neither experimental nor theoretical justification.

## Data Availability

Not applicable.

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
