# Peer review of "The Problem with Inventing Molecular Mechanisms to Fit Thermodynamic Equations of Muscle"

_ijms, 2023, doi:10.3390/ijms242015439_

Round 1

Reviewer 1 Report

First, generally. this should not be a scientific paper of the type methods-data-discussion-conclusions but paper describing author´s view on a specific problem, which is based positively on some historical considerations and author´s previous results. It is headed as “Opinion” and this is exactly what it is. The opinion is rather unconventional, “un-mainstream” but this is just what is necessary in science, its progress, but nowadays not too much published. This is one reason why I would strongly support publishing this manuscript.

Now, specifically. I am not an expert in muscle mechanics etc. but I, hopefully, know something about thermodynamics. Thus, I dare to say that I can provide an unbiased “external” view on the topic. I want to stress several words from the manuscript: “(...)efficient, clean-burning machines(...)” which justify the necessity of thermodynamic view on muscles, the view promoted by this author. The technical details are not, of course, included in such opinion-giving contribution but readers can be encouraged to look at (at least) references given in this manuscript and evaluate this opinion by themselves. Briefly, this opinion is well-written and certain offensive parts are not inappropriate but just more “forcing” or acceptably provocative.

I have some suggestions for improvements. Minor formal comment. I think that references should be extended, especially, when referring to history or quoting directly, for example to lines (and following) 124, 246, 294.

Minor (major?) technical comment. While I do not want to call for extension (the length is adequate to present a opinion, a longer expansion could reduce interest of potential readers) I just suggest to author or editor to consider some additional notes and leave the final decision on them. It seems to me that this text is tightly related to links and differences between micro- and macroscopic description, between molecular and continuum approaches. This aspect could be more commented and considered, cf. also, e.g., Murdoch, A. I. A corpuscular approach to continuum mechanics: Basic considerations. Archive for Rational Mechanics and Analysis. 88(4), 291 (1985).

This opinion is perhaps aimed also at changing current paradigm (at least in muscle models), as I estimate from mentioning T.S.Kuhn. Referring to other philosopher, K.Popper, readers could be informed on comparison of outputs of corpuscular/thermodynamic muscle theories with experimental data as an essential tool to reject soma (any) theory (hypothesis).

Author Response

Response to Reviewer 1 Comments

The comments of Reviewer 1 are much appreciated and have helped to improve this opinion piece. I have responded to all suggestions with revisions to the manuscript.

I have extended the references.

I have toned down some of the more aggressive statements.

I have included the recommended reference in an expanded discussion of corpuscular versus continuum mechanics.

I have added a table and text for a “comparison of outputs” for the two theories.

Reviewer 2 Report

The article: "The Problem with Inventing Molecular Mechanisms to Fit Thermodynamic Equations of Muscle” address the queries concerning two different models of muscle contraction: corpuscular mechanic model (e.g. power stroke) and thermodynamic model (e.g. molecular switch). Huxley’s model assumed that the mechanical properties of a system are determined by the mechanical properties of the molecules within that system, but nowadays association of that theory with microscopic structural details and relation to specific micro-mechanical interactions remain a subject of scientific debate. In thermodynamic model, the conduit for energy transduction is acto-myosin ATPase reaction (not a single myosin motor) within which a molecular switch induced by actin-binding is an intermediate step. The free energy reaction of the system drives the ensemble reaction, and the assembly of molecular switches performs the work through a thermodynamic power stroke.

The author of the manuscript is an enthusiast of the second model and the narrative of this article is conducted in accordance with this model.  Baker presented here the history of both theories and described both, muscle thermodynamics and muscle corpuscular mechanics. Finally, he provided evidence confirming the incompatibility of the two  described models.

In my opinion this text is well written and clear. Whether or not one agree with the author I think his opinion deserves to be presented to the broader scientific community and subjected to deep discussion. However, it is difficult to accept certain expressions that may be perceived as „too strong” for supporters of another theory:

·         Lines 19-20: While its  intuitive appeal is undeniable, corpuscular mechanics is not supported by science.

·         Lines 21-22: With clear implications for human  health, corpuscular mechanic muscle models continue to misinform therapeutic and clinical decision-making - Can the author explain in more detail what he means?

·         Lines 259-260: While its intuitive appeal is undeniable, corpuscular mechanics is not science, and  thus its use as a scientific tool creates a human barrier to scientific progress.

·         Lines 365-366: A  thermodynamic muscle model is based on science; a corpuscular mechanic muscle model  is not

·         Lines 377-378: Throughout the history of science, corpuscular mechanic convictions among scientists have misled science and continue to do so today.

I recommend removing these sentences or toning down their content.

I would suggest preparing a nice figure or table summarizing the main differences between described models.

 It would also be good to discuss the work of: Kimmig et al. Adv. Model. and Simul. in Eng. Sci. (2019) 6:6; https://doi.org/10.1186/s40323-019-0128-9

In lines 63-64 the author wrote: There is no gray  area between these models; What does it mean?

Author Response

Response to Reviewer 2 Comments

I thank Reviewer 2 for the helpful comments. In particular, the table that I have added to the revised manuscript in response to Reviewer 2 recommendations has significantly improved this opinion piece.

I have toned down or removed the expressions that could be perceived as “too strong”.

  • Lines 19-20: While its  intuitive appeal is undeniable, corpuscular mechanics is not supported by science.

       I have removed this statement.

  • Lines 21-22: With clear implications for human  health, corpuscular mechanic muscle models continue to misinform therapeutic and clinical decision-making - Can the author explain in more detail what he means?

       I have revised this statement to make it more clear.

  • Lines 259-260: While its intuitive appeal is undeniable, corpuscular mechanics is not science, and  thus its use as a scientific tool creates a human barrier to scientific progress.

       I have removed this statement.

  • Lines 365-366: A  thermodynamic muscle model is based on science; a corpuscular mechanic muscle model  is not

       I have removed this statement.

  • Lines 377-378: Throughout the history of science, corpuscular mechanic convictions among scientists have misled science and continue to do so today.

I would suggest preparing a nice figure or table summarizing the main differences between described models.

I have prepared and included this table and added text to support it.

 It would also be good to discuss the work of: Kimmig et al. Adv. Model. and Simul. in Eng. Sci. (2019) 6:6; https://doi.org/10.1186/s40323-019-0128-9

I have included a discussion of the work of Kimmig and others. 

In lines 63-64 the author wrote: There is no gray  area between these models; What does it mean?

I have removed “gray area” and have included text that better describes my original intent.